# Dementia and disadvantage in the USA and England: population-based comparative study

Karolos Arapakis ,[1] Eric Brunner ,[2] Eric French,[3,4] Jeremy McCauley [5]

¹Department of Economics, UCL, London, UK
²Department of Epidemiology and Public Health, University College London, London, UK
³Department of Economics, Cambridge University, Cambridge, UK
⁴Institute for Fiscal Studies, London, UK
⁵School of Economics, University of Bristol, Bristol, UK

**Correspondence to**
Dr Jeremy McCauley;
jeremy.mccauley@bristol.ac.uk

## ABSTRACT

**Objectives** To compare dementia prevalence and how it varies by socioeconomic status (SES) across the USA and England.

**Design** Population-based comparative study.

**Setting** Non-Hispanic whites aged over 70 population in the USA and England.

**Participants** Data from the Health and Retirement Study and the English Longitudinal Study of Ageing, which are harmonised, nationally representative panel studies. The sample includes 5330 and 3147 individuals in the USA and England, respectively.

**Main outcome measures** Between country differences in age-gender standardised dementia prevalence, across the SES gradient. Dementia prevalence was estimated in each country using an algorithm based on an identical battery of demographic, cognitive and functional measures.

**Results** Dementia prevalence is higher among the disadvantaged in both countries, with the USA being more unequal according to four measures of SES. Overall prevalence was lower in England at 9.7% (95% CI 8.9% to 10.6%) than the USA at 11.2% (95% CI 10.6% to 11.8%), a difference of 1.4 percentage points (pp) (p=0.0055). Most of the between country difference is driven by the bottom of the SES distribution. In the lowest income decile individuals in the USA had 7.3 pp (p<0.0001) higher prevalence than in England. Once past health factors and education were controlled for, most of the within country inequalities disappeared; however, the cross-country difference in prevalence for those in lowest income decile remained disproportionately high.

**Conclusions** There is inequality in dementia prevalence according to income, wealth and education in both the USA and England. England has lower dementia prevalence and a less steep SES gradient. Most of the cross-country difference is concentrated in the lowest SES group, which provides evidence that disadvantage in the USA is a disproportionately high risk factor for dementia.

## INTRODUCTION

Dementia, a severe and irreversible decline in memory and other cognitive functions, is a major and increasing global health challenge. It is the fifth leading cause of death globally and is one of the most common comorbidities for COVID-19 morbidity.[1 2] It results in large social and economic costs.[3 4]

## Strengths and limitations of this study

⇒ This is the first study to compare dementia prevalence across countries using the same survey methodology and the exact same measure of dementia. The surveys have similar sample selection and questionnaire design. We standardise our estimates by age and gender to the English population aged over 70 in 2016. Any differences in overall prevalence across the two countries should represent true differences.

⇒ We measure the socioeconomic status (SES) gradient of dementia across four different measures of SES: income; education; wealth and non-housing wealth.

⇒ Dementia disproportionately affects the most disadvantaged in both countries, although the gradient is steeper in the USA according to all four measures of SES.

⇒ We do not ascertain dementia directly but predict cases using a common battery of measures in English Longitudinal Study of Ageing and Health and Retirement Study. One of the SES measures, education, is also used as a predictor of dementia.

Americans are more likely to be in poor health than their English counterparts in multiple dimensions, including heart disease and diabetes.[5] These differences are large along all points of the socioeconomic status (SES) gradient, although the gradient is generally steeper in the USA. While the SES gradient for many diseases has been well established,[6–8] only a few studies have focused specifically on dementia.[9 10] The available evidence is summarised in online supplemental table A1. The evidence of the SES gradient for dementia is also less clear, as in England while a strong association has been established between wealth and dementia incidence, the same was not observed for education level.[11]

The Global Burden of Disease Study (GBD) reported that in 2017 among those aged over 70, the USA had a lower overall prevalence of dementia at 7.89% compared with the UK

at 8.91%.[12] However, the GBD has identified substantial heterogeneity in case-ascertainment methods throughout the dementia literature, resulting in location-specific inconsistencies and potentially biased cross-country comparisons. This has led to calls for analyses with more consistent and comparable measures of dementia to inform policy makers, researchers and clinicians about global differences in dementia.[13]

In this study, we compared dementia prevalence in England and the USA among non-Hispanic whites aged 70+, and how it varied across the SES gradient of each country. Location-specific inconsistencies caused by differences in diagnostic practices were not an issue in our study because we used an identical case definition for dementia, and the surveys in the analysis shared the same design and sampling techniques. More specifically, we used two large surveys, the US Health and Retirement Study (HRS) and the English Longitudinal Study of Ageing (ELSA), that contain a battery of the same demographic, cognitive and functional measures, and we applied the same prediction algorithm in both countries to detect undiagnosed as well as diagnosed cases. We compared dementia prevalence within and across England and the USA using important indicators of SES, specifically: income; education; wealth and non-housing wealth.

## METHODS
### Description of surveys
Data were extracted from the 2016 and earlier waves of the HRS and ELSA, which are nationally representative biennial surveys of the USA and English populations, respectively.[14][15] Both the HRS and ELSA follow respondents longitudinally until death, with new cohorts entering to maintain population representativeness as the study sample gets older. The design of ELSA was based on the HRS, making the two surveys analogous, with both collecting data on health, ability, demographics, employment and wealth. In addition to measuring health conditions and difficulties respondents have with Activities of Daily Living (ADLs) and Instrumental Activities of Daily Living (IADLs), sample members also have their cognitive function assessed. A range of tests adapted from the Telephone Interview for Cognitive Status (TICS) have been carried out in HRS since 1996 and ELSA since 2014. If a sample member was unable to respond in person, a proxy respondent was asked questions about the respondent's change in memory. Both surveys have a high response rate, which is displayed in the online supplemental appendix A2. We describe these surveys in more detail in the online supplemental appendix.

### Cohort description
Our samples are restricted to non-Hispanic whites over the age of 70 years old that live in the community or in nursing homes in 2016. This provides a study sample of 5330 participants in the HRS and 3147 participants in

ELSA. We restrict our sample to non-Hispanic whites to ensure estimates are comparable across countries. Summary statistics of both the raw and selected samples are displayed in online supplemental table A3 and A4, and specifications that include ethnic minorities are also displayed.

### Patient and public involvement
No study participants were involved in setting the research question or outcome measures, nor were they involved in any other area of the design, implementation and analysis of the study. There are no direct plans to disseminate the results of the research to study participants.

### Dementia case definition
The HRS included a detailed clinical substudy (ADAMS: Aging, Demographics and Memory Study) of 856 sample members aged 70+ who completed an in-depth in-home assessment of cognitive status conducted by experienced teams at the Duke University Dementia Epidemiology Research Center who diagnosed each participant as normal, cognitively impaired but not demented or demented.[3] Data from ADAMS are regarded as the gold-standard dementia diagnoses against which to train algorithms to predict dementia.[16] Hurd et al estimated separate ordered probit models in the ADAMS subsample for self-respondents and proxy-respondents to generate a predictive algorithm for cognitive status, based on the ADAMS diagnoses, for the whole HRS sample. The algorithm uses a range of variables including demographic information, ADLs, IADLs, TICS questionnaire as well as the change in these variables across waves.[3] Proxy respondents had a separate predictive algorithm as they were asked a different set of questions from self-respondents, which included the short form of the Informant Questionnaire on Cognitive Decline in the Elderly (IQCODE). The use of a proxy to assess cognitive decline and dementia in elderly people is a recognised accepted standard method for identifying severe cognitive impairment and has been validated many times.[17] Importantly, the same set of questions used in the Hurd et al algorithm is asked of self-respondents and proxy-respondents in both HRS and ELSA. Summary statistics for a variety of predictors are displayed in online supplemental table A5.

We applied Hurd et al's predictive algorithm to estimate the probably of dementia for those in the HRS sample in 2016 and extended the prediction to the ELSA sample. The algorithm predicts the probability of dementia in the following year; therefore, we predicted dementia prevalence in 2017. Hurd et al's predictive algorithm has been shown to have an accuracy (percentage correctly classified as demented or non-demented) of 94%, sensitivity of 65% and a specificity of 98% in the estimation sample.[16] An in-depth discussion of the predictive algorithm procedure can be found in the online supplemental appendix.

Non-response for people unable or unwilling to participate in the survey is important when attempting to estimate dementia prevalence across the population. While

attrition exists in both surveys, it is unlikely to significantly affect our estimates since among older ELSA and HRS respondents, there is no statistically significant correlation between attrition and prior health or the SES indicators of education, income and wealth.[18]

## Measures of socioeconomic status

We considered four measures of SES: income; education; wealth and non-housing wealth. Income is measured as current household income from all sources. For education, we used total years of schooling. Wealth is measured as the sum of all household reported savings, stocks, bonds, business wealth, other assets and the value of housing assets (eg, properties) after financial debt and mortgage debt has been subtracted. Non-housing wealth is the same measure as wealth but excludes housing assets and mortgage debt and therefore measures wealth that can be more easily converted to cash. Wealth and non-housing wealth are both measured from 4 years prior to minimise reverse causality, as medical expenses associated with dementia are high and may run down wealth.[4] For each measure, we created a SES gradient by ranking individuals based on that measure. For income, wealth and non-housing wealth, we assigned everyone to a decile in their respective country. For education, we ranked individuals according to their number of years of schooling.

## Statistical analysis

We created a pooled dataset of the two surveys. In our statistical analysis, we used HRS and ELSA sampling weights to adjust for non-response and for the sampling design of the surveys. To make both within country estimates along the SES gradient and cross-country estimates directly comparable, estimates were age-gender standardised to the English population aged over 70 in 2016 using direct standardisation, categorising the population into 10 groups: five age bands (70–74; 75–79; 80–84; 85–89 and 90+) by gender. We estimated the prevalence of dementia in each country, their difference and compared the prevalence along the four SES gradients. For each estimate presented, we computed the corresponding 95% CI, and for any differences we computed the corresponding p values. For each SES factor, as well as estimating the age-gender standardised prevalence along the gradient, we calculated the Relative Index of Inequality (RII) and Slope Index of Inequality (SII) using generalised linear models (log binomial regression) with logarithmic and identity link functions, respectively. The RII can be interpreted as the relative likelihood of dementia prevalence of those in the lowest SES group compared with those in the highest, and the SII can be interpreted as the absolute effect on dementia probability of moving from the lowest SES group to the highest.[19] To assess whether any observed differences could be explained by disparities in past health risk factors across countries, we conditioned on a variety of risk factors and assessed how our estimates changed. Where possible, when conditioning on these factors, we used past health instead of current health

to address the problem of reverse causality: that is, the problem that dementia may cause health problems such as low weight. Statistical analyses were performed using STATA software.

## RESULTS

Table 1 shows the age-gender standardised prevalence of dementia for the aged over 70 white non-Hispanic population in both England and the USA. Dementia prevalence is lower in England at 9.7% (95% CI 8.9% to 10.6%) than the USA at 11.2% (95% CI 10.6% to 11.8), a difference of 1.4 percentage points (pp) that is highly statistically significant (p=0.0055).

Table 1 also shows dementia prevalence for different SES groups, in terms of income, education, wealth and non-housing wealth. Regardless of the measure of the SES, there is a clear gradient in dementia prevalence, with the most disadvantaged groups in both England and the USA having higher dementia prevalence. The gradient is steeper in the USA and is driven by significantly higher dementia prevalence for those at the very bottom of the distribution. In the USA, those in the lowest income decile have a dementia prevalence of 18.7% (95% CI 16.6% to 20.8%), which is considerably higher than in England, with a prevalence among those in the lowest decile of 11.4% (95% CI 8.9% to 13.9%). The difference is highly statistically significant (p<0.0001). For income deciles above the lowest, the difference across the two countries is much smaller and not statistically significant. This same general pattern is evident across the other measures of SES that we consider, although when using wealth, the difference between those in the bottom decile is not statistically significant.

Figure 1 presents the same dementia prevalence information shown in table 1, but in graphical format. It also reports the SII for the four different measures of the SES for both countries. In both the USA and England, dementia is more prevalent among the more disadvantaged. The gradient tends to be steeper in the USA, corresponding to a larger (in absolute value) SII in the USA for each SES measure. For income, the SII is −0.062 (95% CI −0.097 to −0.028) and −0.085 (95% CI −0.114 to −0.057) for England and the USA, respectively. The SIIs are not statistically different. If the lowest income decile is excluded, the SII for England becomes slightly steeper (−0.067 (95% CI −0.107 to −0.027)) whereas the SII for the USA becomes less steep than England (−0.060 (95% CI −0.093 to −0.027)).

Next, we attempted to understand the potential drivers of these gradients and the differences in the gradients across countries. We extended the analysis to account for cardiometabolic diseases (ie, diabetes, heart disease and stroke) and behaviours (ie, smoking and body mass index) as dementia risk factors.[1 20] Previous research showed these factors to be more prevalent in the USA than England, especially among the most disadvantaged.[5] Table 2 displays the percentage point difference

**Table 1** Prevalence of dementia, USA versus England, 2017

| | England | USA | | |
| --- | --- | --- | --- | --- |
| | Age-gender standardised prevalence (95% CI) | Age-gender standardised prevalence (95% CI) | Difference | P value |
| All | 0.097 (0.089 to 0.106) | 0.112 (0.106 to 0.118) | 0.014 | 0.0055 |
| Household income decile | | | | |
| 1 (lowest) | 0.114 (0.089 to 0.139) | 0.187 (0.166 to 0.208) | 0.073 | <0.0001 |
| 2 | 0.113 (0.090 to 0.136) | 0.141 (0.119 to 0.163) | 0.028 | 0.090 |
| 3 | 0.124 (0.097 to 0.151) | 0.111 (0.095 to 0.127) | −0.013 | 0.42 |
| 4 | 0.099 (0.071 to 0.126) | 0.118 (0.099 to 0.137) | 0.019 | 0.26 |
| 5 | 0.094 (0.072 to 0.116) | 0.086 (0.069 to 0.102) | −0.008 | 0.56 |
| 6 | 0.098 (0.070 to 0.127) | 0.108 (0.088 to 0.128) | 0.010 | 0.59 |
| 7 | 0.068 (0.042 to 0.093) | 0.100 (0.078 to 0.122) | 0.032 | 0.060 |
| 8 | 0.083 (0.053 to 0.114) | 0.082 (0.066 to 0.097) | −0.002 | 0.92 |
| 9 | 0.082 (0.041 to 0.122) | 0.093 (0.071 to 0.116) | 0.012 | 0.62 |
| 10 (highest) | 0.059 (0.035 to 0.083) | 0.077 (0.052 to 0.103) | 0.019 | 0.30 |
| Years of schooling | | | | |
| 9 or fewer | 0.128 (0.101 to 0.154) | 0.190 (0.162 to 0.218) | 0.062 | 0.0015 |
| 10 | 0.095 (0.074 to 0.116) | 0.137 (0.109 to 0.165) | 0.042 | 0.018 |
| 11 | 0.096 (0.077 to 0.115) | 0.109 (0.080 to 0.139) | 0.013 | 0.471 |
| 12 | 0.071 (0.042 to 0.100) | 0.124 (0.114 to 0.133) | 0.053 | 0.0006 |
| 13 | 0.061 (0.038 to 0.083) | 0.116 (0.090 to 0.141) | 0.055 | 0.0013 |
| 14 or more | 0.056 (0.039 to 0.073) | 0.085 (0.076 to 0.093) | 0.029 | 0.0031 |
| Household wealth decile | | | | |
| 1 (lowest) | 0.165 (0.132 to 0.198) | 0.187 (0.162 to 0.211) | 0.022 | 0.31 |
| 2 | 0.117 (0.092 to 0.143) | 0.149 (0.129 to 0.169) | 0.031 | 0.061 |
| 3 | 0.100 (0.073 to 0.127) | 0.107 (0.091 to 0.122) | 0.006 | 0.68 |
| 4 | 0.093 (0.071 to 0.115) | 0.115 (0.098 to 0.132) | 0.022 | 0.12 |
| 5 | 0.110 (0.085 to 0.134) | 0.091 (0.077 to 0.106) | −0.018 | 0.21 |
| 6 | 0.080 (0.057 to 0.103) | 0.089 (0.074 to 0.103) | 0.008 | 0.55 |
| 7 | 0.070 (0.046 to 0.094) | 0.103 (0.084 to 0.123) | 0.034 | 0.034 |
| 8 | 0.092 (0.066 to 0.118) | 0.089 (0.072 to 0.106) | −0.003 | 0.85 |
| 9 | 0.082 (0.048 to 0.116) | 0.103 (0.084 to 0.123) | 0.021 | 0.28 |
| 10 (highest) | 0.060 (0.038 to 0.081) | 0.067 (0.051 to 0.084) | 0.008 | 0.58 |
| Household non-housing wealth decile | | | | |
| 1 (lowest) | 0.136 (0.103 to 0.168) | 0.201 (0.176 to 0.226) | 0.065 | 0.0019 |
| 2 | 0.109 (0.084 to 0.134) | 0.137 (0.119 to 0.156) | 0.029 | 0.074 |
| 3 | 0.123 (0.095 to 0.151) | 0.131 (0.111 to 0.150) | 0.008 | 0.66 |
| 4 | 0.096 (0.073 to 0.118) | 0.101 (0.085 to 0.116) | 0.005 | 0.71 |
| 5 | 0.108 (0.082 to 0.134) | 0.107 (0.091 to 0.123) | −0.001 | 0.95 |
| 6 | 0.093 (0.067 to 0.119) | 0.086 (0.070 to 0.101) | −0.007 | 0.64 |
| 7 | 0.099 (0.072 to 0.125) | 0.086 (0.070 to 0.102) | −0.013 | 0.42 |
| 8 | 0.078 (0.051 to 0.105) | 0.090 (0.073 to 0.106) | 0.011 | 0.49 |
| 9 | 0.058 (0.036 to 0.081) | 0.092 (0.074 to 0.111) | 0.034 | 0.024 |
| 10 (highest) | 0.063 (0.043 to 0.084) | 0.079 (0.062 to 0.095) | 0.015 | 0.26 |

Sample includes non-Hispanic white population aged 70+ only. The sample size is 3147 participants in England and 5330 participants in the USA. All estimates are age-gender standardised to the overall 2016 aged 70+ white population in England. The difference is calculated as the prevalence in the USA minus prevalence in England. All deciles are calculated within country.
Overall prevalence and prevalence according to four measures of socioeconomic status.

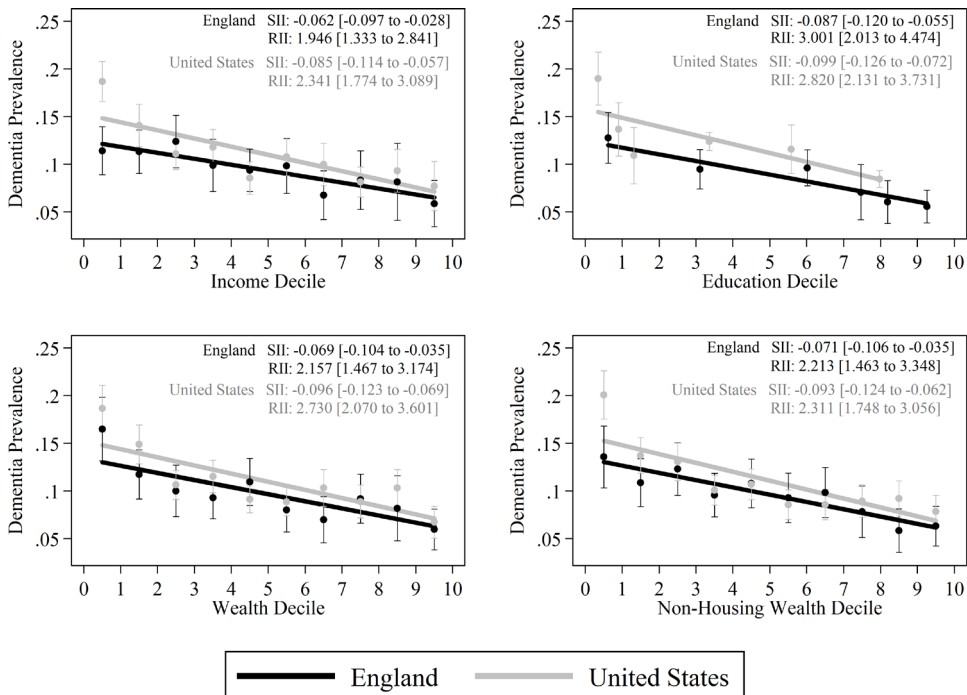

**Figure 1** SES Gradient of Dementia, USA versus England, 2017, according to four measures of SES. Absolute and relative inequality shown with 95% CI. The points in this figure represent the mean age-gender standardised dementia prevalence for each country by SES, along with 95% CI for these predictions. The solid lines represent the fitted Slope Index of Inequality (SII: absolute inequality) for each country. The values of the SII and the corresponding Relative Index of Inequality (RII: relative inequality) are listed in the top right of each figure, with 95% CI in brackets. For education, individuals are ranked based on their years of schooling within each country, and as the USA has higher educational attainment, individuals with 14+ years of schooling are the 80th percentile of the US education distribution, but at the 90th percentile of the English educational distribution. SES, socioeconomic status.

in dementia prevalence after we controlled for various measures of past health and behaviours. The results are split into three panels: the whole sample; the whole sample excluding those in the lowest income decile and the lowest income decile. As was shown in table 1, the difference in the prevalence of dementia between England and the USA was 1.43 pp (p=0.0055). Table 2 shows this difference declined to 0.894 pp (p=0.11) when we excluded the lowest income decile. Controlling for past health and behaviours modestly reduced this cross-country difference further: the difference declined by a maximum of 19%. For the lowest income decile, controlling for past health and behaviour reduced the cross-country difference of 7.27 pp (p<0.0001) by a more substantial 33%.

In the online supplemental tables A6–A8, we investigated whether past health and behaviour explained the SES gradient within each country. We found that in England and the USA these factors accounted for most of the SES gradient, as shown in online supplemental figure A1. However, in the USA, prevalence in the lowest income decile remained disproportionately high.

Education has also been shown to be a risk factor for dementia. Table 1 shows that in both the USA and England, the less educated have higher dementia prevalence. Controlling for education increased the estimated difference across countries, from 1.43 pp to 2.82 pp, as can be seen in the online supplemental table A7. Education

cannot explain these differences since the English have lower educational attainment.

In our main analysis, we exclude ethnic minorities, who have higher dementia prevalence and comprise a larger share of the USA than the English population. Including minorities increased the estimated difference across countries, from 1.43 pp to 2.42 pp, as can be seen in the online supplemental tables A9 and A10. This is largely caused by the high prevalence of dementia among minorities in the USA as displayed in online supplemental table A11.

## DISCUSSION
### Main findings
Using nationally representative samples of older individuals from England and the USA and applying the same algorithmic procedure to predict dementia in both samples, we showed that in both the USA and England dementia is more prevalent among the disadvantaged, and the SES gradient of dementia is steeper in the USA. The steeper gradient in the USA is largely driven by those in the lowest decile. In both countries, most of the SES gradient disappeared when we controlled for past health related factors, although prevalence for those in lowest income decile in the USA remained disproportionately high. If the lowest income decile is excluded from our sample the difference in dementia prevalence across the

**Table 2** Difference in prevalence of dementia, USA vs England, 2017

| Whole sample | | | | | |
|---|---|---|---|---|---|
| Percentage point difference | 1.43 | 1.34 | 1.15 | 1.21 | 1.18 |
| p value | 0.0055 | 0.0091 | 0.025 | 0.020 | 0.034 |
| % Difference from baseline | – | –6% | –19% | –16% | –17% |
| Excluding lowest income decile | | | | | |
| Percentage point difference | 0.89 | 0.88 | 0.74 | 0.75 | 0.81 |
| p value | 0.11 | 0.11 | 0.18 | 0.18 | 0.18 |
| % difference from baseline | – | –2% | –17% | –16% | –10% |
| Lowest income decile | | | | | |
| Percentage point difference | 7.27 | 6.14 | 5.45 | 5.93 | 4.85 |
| p value | <0.0001 | 0.0003 | 0.002 | 0.0011 | 0.012 |
| % difference from baseline | – | –15% | –25% | –18% | –33% |
| Control for | | | | | |
| Past cardiometabolic diseases | | ✓ | ✓ | ✓ | ✓ |
| Past psychiatric conditions | | | ✓ | ✓ | ✓ |
| Ever smoked | | | | ✓ | ✓ |
| Past BMI | | | | | ✓ |

Sample includes non-Hispanic white population aged 70+ only. The sample size is 3147 participants in England and 5330 participants in the USA. All estimates are age-gender standardised to the overall 2016 aged 70+ white population in England. The difference is calculated as the prevalence in the USA minus prevalence in England. Differences are displayed as percentage points. 'Past Cardiometabolic Diseases' and 'Past Psychiatric Conditions' control for whether an individual says they had the conditions 4 years prior. 'Cardiometabolic Diseases' include diabetes, heart disease and stroke. 'Smoking' controls for whether an individual has ever smoked. 'Past BMI' includes dummy variables to control for whether an individual is classed as underweight, *overweight* or *obese*. BMI values are based on when an individual first entered the survey, which is at least 10 years prior.
Estimates of the between-country difference taking account of antecedent health status, smoking and degree of adiposity.
BMI, body mass index.

countries is statistically insignificant, and the remaining SES gradient of dementia is remarkably similar across countries.

While poorer individuals face a higher burden of dementia in both England and the USA, the extremely poor in the USA face a disproportionately high burden of dementia. Controlling for past health-related factors can explain some, but not all, of the cross-country difference. It can explain up to around one third of the difference for those in the lowest income decile. While past health factors such as adiposity and smoking are correlated with dementia, those in the lowest income decile in the USA do not smoke more or have higher body mass indexes (BMIs) than their English counterparts. Therefore, this cannot explain their disproportionately higher prevalence of dementia. Education also cannot explain the difference, as the US population is more educated at every income decile, and in fact the educational difference masks some of the underlying dementia risk difference between countries.

Adding minorities increased estimated dementia prevalence, especially in the USA, because dementia prevalence is higher among minorities, who comprise a higher share of the USA than the English population. This fits with prior research which showed dementia prevalence is higher for non-whites and Hispanics.[20 21] We did not observe higher dementia prevalence among Hispanics and non-whites in the USA for those in relatively high socioeconomic groups.

### Comparison with previous studies and how findings are an advance on current literature

Previous studies have shown cross-country variation in dementia prevalence. However, substantial heterogeneity in case-ascertainment methods across countries and studies makes interpreting any observed differences difficult. We believe this is the first study to compare dementia prevalence in England and the USA using the exact same measure of dementia, thus overcoming previous difficulties in making comparisons across the two countries due differences in diagnostic practices and case definitions. We also compared the SES gradient of dementia in both countries. While some studies have shown in both England and the USA those with lower education and less wealth have been found to have higher rates of dementia,[4 9 11 22 23] there are no systematic comparative studies. We compared prevalence along the SES gradient using almost identical measures of income, wealth and education. Further, we standardised the cross-country comparison for age and gender, using the English over 70 population as the standard population. We found that dementia prevalence is higher and more concentrated

among the poorest in the USA than England. Detailed disaggregation according to SES measures shows the true extent of the excess burden of dementia in the very poorest group in the USA.

We showed that risk factors for dementia such as cardiometabolic diseases, psychiatric conditions, high BMI, smoking have similar affects across countries. Accounting for these risk factors removes most of the SES gradient for both countries, but disproportionately high prevalence remains for the most disadvantaged in the USA.

### Implications (wider interpretation)

Much research has shown that low-income Americans are more likely to be in poor health and die younger than their high-income counterparts.[24] We show that these health differences also extend to dementia prevalence.

While risk factors contribute to higher prevalence among those who are more disadvantaged, those in the USA appear to have an undue burden that is caused by risk factors for which we cannot account. One possible explanation is differential access to healthcare. The NHS provides broadly equitable care according to education in the older population after accounting for health status.[25] In the USA, the poor often go uninsured, and although virtually every American aged 65 or older is eligible for Medicare, around 20% of Medicare beneficiaries healthcare must be financed out of pocket.[26] The extent to which healthcare provision below and above aged 65 may account for the relative excess dementia burden in the USA is unclear.

The implications of our results are that interventions designed to attempt to prevent dementia should be targeted towards the most disadvantaged. This is especially true in the USA. As yet, we are unable to advocate specific measures as we do not yet understand the specific nature of disadvantage in respect to dementia risk.

### Strengths and weaknesses of analysis

This study has a strong design. Results are directly comparable across England and the USA. The same predictive algorithm was applied to both countries, addressing the problem of heterogeneity in case ascertainment which has affected the literature.[1 13] Further, because ELSA and HRS share sample selection and questionnaire design, any differences in overall prevalence and SES gradients in prevalence across the two countries should represent true differences. In contrast to the Global Burden Disease study, we find higher dementia prevalence in the USA. Furthermore, we measured the SES gradient of dementia across four different measures of SES, with consistent results. Our work also highlights the usefulness of the standardised measure of dementia to allow for meaningful comparisons across countries.

This study had three limitations. First, we do not ascertain dementia directly, but predict cases using a common battery of measures in ELSA and HRS. Importantly, Hurd's prediction algorithm has high accuracy and although our case definition lacks a clinical point of reference in England, it is based on a detailed clinical substudy in the USA.[16] Further, cross-cultural subjectivities in reporting of impairment severity are likely to be similar in the USA and England (see online supplemental table A5). It would be of great value for future work to use the Harmonized Cognitive Assessment Protocol data to provide a standard clinical point of reference to validate and verify the cross-country dementia prevalence estimates.[27] Second, education is one of the factors in the predictive algorithm for dementia and also one of our measures of SES. The dementia algorithm takes account of the well-documented correlation between education level and cognitive function in adult life. Nevertheless, we found substantial absolute and relative inequalities in dementia prevalence according to education level in the USA and UK. Education may be a successful approach for reducing dementia risk.[28] Third, while we show risk factors explain a large proportion of the differences in dementia between the England and the USA—although cannot account for the difference in the lowest income decile—there are likely other unmeasured confounding factors that impact dementia prevalence which we do not observe.

## CONCLUSION

Given the large social and economic costs of dementia, there is great value in understanding the scope and burden of dementia in the population along the SES gradient. This study indicates that more disadvantaged individuals face a higher burden of dementia and that the poorest individuals in the USA face a disproportionately high burden. The high burden faced by these individuals can be partly but not fully explained by past health factors. We lacked data on other important possible contributing factors such as habitual drug use. Further research is needed to fully understand this issue using data from multiple sources.

**Correction notice** The article has been corrected since it was published online. The funding statement has been updated to 'This work was supported by grants from the British Heart Foundation (RG/13/2/30098, RG/16/11/32334), Michigan Retirement Research Center, ESRC (RES-544-28-50001, ES/P001831/1), and the Alexander S. Onassis Foundation.'

**Contributors** All authors made a substantial contribution to study conception and design. EF and JM developed the original idea. JM and KA prepared and analysed the data, with inputs from EF and EB. All authors contributed to interpreting the results and drafting the manuscript. JM and KA serve as guarantors and affirm that the manuscript is an honest, accurate and transparent account of the study being reported; that no important aspects of the study have been omitted and that any discrepancies from the study as planned have been explained. All authors had full access to all of the data (including statistical reports and tables) in the study and take responsibility for the integrity of the data and accuracy of the data analysis. criteria and that no others meeting the criteria have been omitted.

**Funding** This work was supported by grants from the British Heart Foundation (RG/13/2/30098, RG/16/11/32334), Michigan Retirement Research Center, ESRC (RES-544-28-50001, ES/P001831/1), and the Alexander S. Onassis Foundation.

**Competing interests** None declared.

**Patient consent for publication** Not required.

**Provenance and peer review** Not commissioned; externally peer reviewed.

**Data availability statement** Data are available in a public, open access repository. Appendix with details on the formula and calculations is available online. Code and results are available on request from the corresponding author. Appendix with details on the formula and calculations is available online. Code and results are available on request from the corresponding author.

**ORCID iDs**
Karolos Arapakis http://orcid.org/0000-0001-5690-5472
Eric Brunner http://orcid.org/0000-0002-0595-4474
Jeremy McCauley http://orcid.org/0000-0002-8402-483X

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
