## [Reviewer comments · BMJ Open]

ARTICLE DETAILS

TITLE (PROVISIONAL)	Dementia and Disadvantage in the United States and England: Population-based comparative study
AUTHORS	Arapakis, Karolos; Brunner, Eric; French, Eric; McCauley, Jeremy

VERSION 1 – REVIEW

REVIEWER	Ohara, Tomoyuki Kyushu University, Neuropsychiatry
REVIEW RETURNED	30-Dec-2020

GENERAL COMMENTS	This paper compared the prevalence of dementia and its socioeconomic status between England and United States by using the exact same measurement. The objective of this study is interesting and important. However, there are some issues to be concerned. 1. In the Methods section, the authors mentioned that the study design of ELSA was based on the HRS. Although there is a detailed description of these two studies in Data Appendix, the authors should cite the relevant papers.2. The authors commented that study population was restricted to non-Hispanic whites aged 70 years or older in the community or in nursing homes in 2016 to compare the prevalence of dementia and its socioeconomic status between England and United States. Although Table A3 shows an age- and gender-standardized summary of the study participants between the countries, there is no information of the crude characteristics including its response rate. Did the authors assess the distribution of age, sex, living place across the countries? In the prevalence study, the baseline characteristics of study participants including participation rate and its response rate is very important to compare the prevalence rate between the studies.3. There is no information about the response and non-response rate between the studies. Was there a difference of the non-response rate between the studies?4. Hurd et al's algorithm to estimate having dementia showed the high accuracy rate of 93%. Do this mean a sensitivity rate? If so, the authors should show its specificity or discrimination rate.
--

	5. The authors measured the wealth and non-housing wealth. I cannot understand what non-housing wealth means. Please add detailed information about non-housing wealth. 6. In the statistical analysis, the authors commented that “we used the respondent’s inverse probability of being sampled to adjust for nonresponse and the sampling design of the surveys”. I wonder this method is appropriate when the participants’ response rate was significantly higher than that of non-response rate. Please certify the validity of this methods. 7. The authors conducted the age- and gender-standardized estimation by using direct methods. Please clarify the standard population the authors used. In addition, how did the authors standardize the gender? 8. In the Discussion section, the authors comment this study is the first to compare dementia prevalence in England and the US. I am afraid such claims may offend authors whose earlier papers on the topic may have appeared elsewhere. 9. In the limitation section, there must be unmeasured confounding factors in this study.
--	--

REVIEWER	Raina, Sunil Kumar Dr Rajendra Prasad Government Medical College, Community Medicine
REVIEW RETURNED	21-Jan-2021

GENERAL COMMENTS	1) One of the concerns is that a proxy respondent’s response was taken into account in situations where the case/respondent himself was not available in person. This could result in information bias as proxy respondent may not give accurate information regarding the case/respondent himself. 2) Methodology needs to be explained in detail. The process of data collection followed in ELSA and HRS has not been mentioned and if there is a difference in the process of two, that could account for limitation in this study. 3) Another issue was that how did they addressed the location specific inconsistencies in burden of dementia. 4) Lastly what new can be derived from this study was not clear. They mentioned that previously two studies had been done in US and UK which addressed the same question, however what new they were going to derive from this present study was not clearly mentioned in the manuscript.
---

VERSION 1 – AUTHOR RESPONSE

Reviewer: 1 (Dr. Tomoyuki Ohara, Kyushu University)

1. In the Methods section, the authors mentioned that the study design of ELSA was based on the HRS. Although there is a detailed description of these two studies in Data Appendix, the authors should cite the relevant papers.

OUR RESPONSE >> This is very a useful suggestion that we have followed. We have expanded our description of the surveys in the main manuscript (page 3):

“Data were extracted from the 2016 and earlier waves of the HRS and ELSA, which are nationally representative biennial surveys of the US and English populations, respectively. Both the HRS and ELSA follow respondents longitudinally until death, with new cohorts entering to maintain population representativeness as the study sample gets older. The design of ELSA was based on the HRS, making the two surveys analogous, with both collecting data on health, cognitive functioning, demographics, employment, and wealth. In addition to detailing health conditions and difficulties respondents have with Activities of Daily Living (ADLs) and Instrumental Activities of Daily Living (IADLs), sample members also have their cognitive functioning assessed. A range of tests adapted from the Telephone Interview for Cognitive Status (TICS) have been carried out in HRS since 1996 and ELSA since 2014. If a sample member was unable to respond in person, a proxy respondent was asked questions about the respondent’s change in memory. Both surveys have a high response rate, which is displayed in the online appendix Table A2. We describe these surveys in more detail in the online appendix.”

In addition, we now cite:

Sonnega A, Faul JD, Ofstedal MB, Langa KM, Phillips JW, Weir DR. Cohort Profile: the Health and Retirement Study (HRS). *Int J Epidemiol.* 2014 Apr;43(2):576-85. doi: 10.1093/ije/dyu067. Epub 2014 Mar 25. PMID: 24671021; PMCID: PMC3997380.

Step toe A, Breeze E, Banks J, Nazroo J. Cohort profile: the English longitudinal study of ageing. *Int J Epidemiol.* 2013 Dec;42(6):1640-8. doi: 10.1093/ije/dys168. Epub 2012 Nov 9. PMID: 23143611; PMCID: PMC3900867.

2. The authors commented that study population was restricted to non-Hispanic whites aged 70 years or older in the community or in nursing homes in 2016 to compare the prevalence of dementia and its socioeconomic status between England and United States. Although Table A3 shows an age- and gender standardized summary of the study participants between the countries, there is no information of the crude characteristics including its response rate. Did the authors assess the distribution of age, sex, living place across the countries? In the prevalence study, the baseline characteristics of study participants including participation rate and its response rate is very important to compare the prevalence rate between the studies.

OUR RESPONSE >> Thank you for this point, we have added more details regarding the response rates and characteristics of our samples as outlined below.

We have added a breakdown of the response rate to both the surveys (HRS and ELSA) in appendix Table A2. Both surveys have a high response rate. Importantly for our study, Banks et al. (2011) showed that attrition in HRS and ELSA for older respondents is not linked to education, income and/or wealth. We have added detailed summary statistics of the characteristics of our raw samples. Table A3 shows the raw summary statistics for the over 70 non-Hispanic white sample populations in both HRS and ELSA. Included in this table is also the summary statistics of the sample when we used surveys sample weights to adjust for non-response. Reassuringly, when sample weights are applied, the mean characteristics

remain similar, which provides confidence that our samples are highly representative of the populations from which they are drawn.

3. There is no information about the response and non-response rate between the studies. Was there a difference of the non-response rate between the studies?

OUR RESPONSE >> We thank the referee for this comment. We did not previously address the importance of response rates (both survey and item non-response) and the potential for biases. We have added a detailed breakdown of the response rate to appendix Table A2. Both studies have a high response rate, and reassuringly when we applied survey sample weights to adjust for non-response, the mean characteristics remained similar. Item non-response is not an issue in our study as the prediction algorithm we used includes dummy variables for whether the respondent did not answer certain questions, and this factored into the prediction of whether they are cognitively impaired.

4. Hurd et al's algorithm to estimate having dementia showed the high accuracy rate of 93%. Do this mean a sensitivity rate? If so, the authors should show its specificity or discrimination rate.

OUR RESPONSE >> The accuracy rate reported is the percentage correctly classified (as demented or non-demented) and is therefore a mixture of the sensitivity and specificity rates. In Gianattasio et al. (2019) the Hurd et al. algorithm is shown to have an accuracy of 94%, sensitivity of 65%, and a specificity of 98% in the estimation sample. Furthermore, the algorithm is shown to perform well compared to other dementia prediction algorithms. We now describe accuracy and have added the sensitivity and specificity from Gianattasio et al. (2019) into the manuscript on page 4.

5. The authors measured the wealth and non-housing wealth. I cannot understand what non-housing wealth means. Please add detailed information about non-housing wealth.

OUR RESPONSE >> Thank you for pointing out this oversight; we failed to define these variables adequately in the manuscript. We have added the following definitions into the manuscript (page 5): "Wealth is measured as the sum of all household reported savings, stocks, bonds, business wealth, other assets, and the value of housing assets (e.g., properties) after financial debt and mortgage debt has been subtracted. Non-housing wealth is the same measure as wealth but excludes housing assets and mortgage debt, and therefore gives a better measure wealth that can be more easily converted to cash."

6. In the statistical analysis, the authors commented that "we used the respondent's inverse probability of being sampled to adjust for nonresponse and the sampling design of the surveys". I wonder this method is appropriate when the participants' response rate was significantly higher than that of non-response rate. Please certify the validity of this methods.

OUR RESPONSE >> Thank you for this question, we apologise for being unclear in our wording and appreciate this likely caused confusion. We have now re-worded this sentence to make clear we are using a standard approach of sample weights to adjust for non-response and survey design. We now say (page 5):

"We used HRS and ELSA sampling weights to adjust for nonresponse and for the sampling design of the surveys."

7. The authors conducted the age- and gender-standardized estimation by using direct methods. Please clarify the standard population the authors used. In addition, how did the authors standardize the gender?

OUR RESPONSE >> We use the English population aged over 70 in 2016 as the standard population, and use direct standardization (dstsize command in Stata). On page 5 the following text was added "categorising the population into ten groups: across five age bands (70-74; 75-79; 80-84; 85-89; and 90+) by gender".

We then calculate the conditional mean prevalence for each of these groups by socioeconomic decile and use the proportions of the population of these groups in England (2016) to weight the conditional means.

8. In the Discussion section, the authors comment this study is the first to compare dementia prevalence in England and the US. I am afraid such claims may offend authors whose earlier papers on the topic may have appeared elsewhere.

OUR RESPONSE >> Thank you for this comment; we were previously unclear on our contribution relative to others, which we have amended on page 7 of the manuscript.

We discuss previous studies (including the Global Burden of Disease study) throughout the paper and are indebted to their work comparing dementia across countries. Our study builds on this work by comparing dementia prevalence across the US and England using the same survey methodology and the exact same measure of dementia. This improves on previous studies who have differences in case definitions across countries. We have also been careful to standardize our analysis to the English population according to age and gender structure. Therefore, we believe any differences across countries should represent true differences. While we recognise this is incremental progress, we believe our study is novel in this respect and our methodology provides contrasting results to previous studies. For example, the Global Burden Disease study finds higher dementia prevalence in England than the US, whereas we find higher dementia prevalence in the US than England. This difference is due to the methodologies, of which we believe ours is an improved approach. We also present some striking new findings regarding the socioeconomic gradient of dementia between the US and England and we show the true extent of how disproportionately dementia affects the most disadvantaged in the US.

9. In the limitation section, there must be unmeasured confounding factors in this study.

OUR RESPONSE >> This is a valid point and we responded by adding the following text into the Limitations section of the manuscript (page 9):

“... while we show risk factors explain a large proportion of the differences in dementia between the England and the US – although cannot account for the difference in the lowest income decile – there are likely other unmeasured confounding factors that impact dementia prevalence which we do not observe.”

Reviewer: 2 (Dr. Sunil Kumar Raina, Dr Rajendra Prasad Government Medical College, Dr Rajendra Prasad Government Medical College)

1. One of the concerns is that a proxy respondent's response was taken into account in situations where the case/respondent himself was not available in person. This could result in information bias as proxy respondent may not give accurate information regarding the case/respondent himself.

OUR RESPONSE >> This is a valid methodological concern; however, we believe this issue is accounted for in our study. The Hurd et al. algorithm applies a separate ordered probit model to proxy-respondents for this very concern, so that proxy respondents have their own prediction algorithm that is separate from the self-respondents' algorithm. Proxy respondents were asked a different set of questions from self-respondents, which included the short form of the Informant Questionnaire on Cognitive Decline in the Elderly (IQCODE). The use of a proxy/relative to assess cognitive decline and dementia in elderly people is a recognised accepted standard method for identifying severe cognitive impairment and has been validated many times (see Jorm et al., 2000). Importantly for us, the questions asked to the proxy are the same in both HRS (US) and ELSA (England) and therefore using the proxy algorithm to diagnose cognitive decline created by Hurd et al. is applied in the same fashion as the self-respondent algorithm. We now make this point clear in our section on "Dementia Case Definition" found on page 4 of the manuscript.

2. Methodology needs to be explained in detail. The process of data collection followed in ELSA and HRS has not been mentioned and if there is a difference in the process of two, that could account for limitation in this study.

OUR RESPONSE >> Thank you for this point, we recognise we did not explain the surveys in enough detail. We have added more detail in our description of both the HRS and ELSA by including a subsection "Description of Surveys" on page 3 of the manuscript. Please also refer to response to Reviewer 1 points 1, 2 and 3. One of the main advantages of our study is the fact that the ELSA survey design is based on HRS and as such they are highly comparable. We have also expanded the description of the methodology of how we predict dementia cases in both surveys, using the same prediction algorithm based on clinical diagnoses of a subsample from the HRS (page 4). Crucially, we are using identical measures in HRS and ELSA to predict dementia cases.

3. Another issue was that how did they address the location specific inconsistencies in burden of dementia.

OUR RESPONSE >> Thank you for this comment. We believe that you are concerned about differences across the two countries in the definition of dementia and differences in response rates across the two countries. In the introduction we previously alluded to this issue, as we believe this is the main contribution of the study, but now we can see that we were unclear. We believe the amended introduction is now much clearer. In particular, we write (page 3):

"Location-specific inconsistencies caused by differences in diagnostic practices were not an issue in our study because we used an identical case definition for dementia, and the surveys in the analysis shared the same design and sampling techniques. More specifically, we used two large surveys, the US Health and Retirement Study (HRS) and the English Longitudinal Study of Ageing (ELSA), that contain a battery of the same demographic, cognitive, and functional measures, and we applied the same prediction algorithm in both countries to detect undiagnosed as well as diagnosed cases."

We now more clearly explain how we define dementia in the two countries and have extended our description of the Hurd et al. dementia prediction algorithm in our methodology (page 4):

"The HRS included a detailed clinical substudy (ADAMS: Aging, Demographics and Memory Study) of 856 sample members aged 70+ who completed an in-depth in-home assessment of cognitive status conducted by experienced teams at the Duke University Dementia Epidemiology Research Center who diagnosed each participant as normal, cognitively impaired but not demented (CIND), or demented. Data from ADAMS is regarded as the gold-standard dementia diagnoses against which to train algorithms to predict dementia. Hurd et al. estimated separate ordered probit models in the ADAMS subsample for self- and proxy-respondents to generate a predictive algorithm for cognitive status, based on the ADAMS diagnoses, for the whole HRS sample. The algorithm uses a range of variables including demographic information, Activities of Daily Living (ADLs), Instrumental Activities of Daily Living (IADLs), TICS questionnaire, as well as the change in these variables across waves...."

...We applied Hurd et al.'s predictive algorithm to estimate the probability of dementia for those in the HRS sample in 2016 and extended the prediction to the ELSA sample."

We also now explain more clearly how the HRS and ELSA surveys are consistent in their design and questions (page 3). Response rates across the two surveys are now listed in Table A2. Both surveys have a high response rate and, importantly for our study, Banks et al. (2011) show that attrition in HRS and ELSA for older respondents is not linked to education, income and/or wealth.

Further, we standardise our estimates by age and gender to the English population aged over 70 in 2016. Please see response to Reviewer 1 point 7. Given that we use the exact same definition of dementia in both countries and we have standardized the populations, we believe that any differences in dementia across countries should represent true differences, with no location specific inconsistencies.

4. Lastly what new can be derived from this study was not clear. They mentioned that previously two studies had been done in US and UK which addressed the same question, however what new they were going to derive from this present study was not clearly mentioned in the manuscript.

OUR RESPONSE >> Thank you for pointing this out - we now see that we were unclear in highlighting the key contribution of the paper. We have amended the section "Comparison with previous studies and how findings are an advance on current literature" with the following text (page 7):

"Previous studies have shown cross-country variation in dementia prevalence, however substantial heterogeneity in case-ascertainment methods across countries and studies makes interpreting any observed differences difficult. We believe this is the first study to compare dementia prevalence in England and the US using the exact same measure of dementia, thus overcoming previous difficulties in making comparisons across the two countries due differences in diagnostic practices and case definitions. We also compared the SES gradient of dementia in both countries. While, some studies have shown in both England and the US those with lower education and less wealth have been found to have higher rates of dementia, there are no systematic comparative studies. We compared prevalence along the SES gradient using almost identical measures of income, wealth, and education. Further, we standardized the cross-country comparison for age and gender, using the English over 70 population as the standard population. We found that dementia prevalence is higher and more concentrated among the poorest in the US than England. Detailed disaggregation according to SES measures shows the true extent of the excess burden of dementia in the very poorest group in the US."

Our methodology provides contrasting results to previous studies. For example, the Global Burden Disease study finds higher dementia prevalence in England than the US, whereas we find higher dementia prevalence in the US than England. This difference is due to the methodologies, of which we believe ours is an improved approach.

VERSION 2 – REVIEW

REVIEWER	Ohara, Tomoyuki Kyushu University, Neuropsychiatry
REVIEW RETURNED	20-Mar-2021
GENERAL COMMENTS	The authors have satisfactorily responded to all my questions and made the necessary changes to the manuscript.

VERSION 2 – AUTHOR RESPONSE

We believe we have every element of the STROBE checklist already accounted for in our paper and have therefore made no new changes from the previous submission.